# Effect of Nano Silica Particles on Impact Resistance and Durability of Concrete Containing Coal Fly Ash

**DOI:** 10.3390/nano11051296

**Published:** 2021-05-14

**Authors:** Peng Zhang, Dehao Sha, Qingfu Li, Shikun Zhao, Yifeng Ling

**Affiliations:** 1School of Water Conservancy Engineering, Zhengzhou University, Zhengzhou 450001, China; zhangpeng@zzu.edu.cn (P.Z.); a18637195898@163.com (D.S.); skzhaozzu@163.com (S.Z.); 2Department of Civil, Construction and Environmental Engineering, Iowa State University, Ames, IA 50011, USA; yling@iastate.edu

**Keywords:** concrete, NS, mechanical properties, impact resistance, durability

## Abstract

In this study, the effect of adding nano-silica (NS) particles on the properties of concrete containing coal fly ash were explored, including the mechanical properties, impact resistance, chloride penetration resistance, and freezing–thawing resistance. The NS particles were added into the concrete at 1%, 2%, 3%, 4%, and 5% of the binder weight. The behavior under an impact load was measured using a drop weight impact method, and the number of blows and impact energy difference was used to assess the impact resistance of the specimens. The durability of the concrete includes its chloride penetration and freezing–thawing resistance; these were calculated based on the chloride diffusion coefficient and relative dynamic elastic modulus (RDEM) of the samples after the freezing–thawing cycles, respectively. The experimental results showed that the addition of NS can considerably improve the mechanical properties of concrete, along with its freezing–thawing resistance and chloride penetration resistance. When NS particles were added at different replacement levels, the compressive, flexural, and splitting tensile strengths of the specimens were increased by 15.5%, 27.3%, and 19%, respectively, as compared with a control concrete. The addition of NS enhanced the impact resistance of the concrete, although the brittleness characteristics of the concrete did not change. When the content of the NS particles was 2%, the number of first crack impacts reached a maximum of 37, 23.3% higher compared with the control concrete. Simultaneously, the chloride penetration resistance and freezing–thawing resistance of the samples increased dramatically. The optimal level of cement replacement by NS in concrete for achieving the best impact resistance and durability was 2–3 wt%. It was found that when the percentage of the NS in the cement paste was excessively high, the improvement from adding NS to the properties of the concrete were reduced, and could even lead to negative impacts on the impact resistance and durability of the concrete.

## 1. Introduction

Owing to its high strength, extra workability, excellent hydrothermal stability, and abundant raw material resources for easy mixing, normal concrete has become the building material with the most frequent use and widest application range [1]. However, as the scope of concrete applications becomes wider, higher requirements for concrete properties (such as its durability and impact resistance) are being proposed, e.g., for high-rise and large-span buildings, building structures in severely cold areas, and other special purposes [2,3]. Therefore, the inclusion of different types of admixtures (such as various nanomaterials and fibers) in cement-based materials has become a common method for researchers to improve the properties of concrete [4,5,6]. Nanomaterials are materials comprising of particles with particle sizes between 1–100 nm. Owing to the ultrafine sizes of the nanoparticles, the surface electronic structures and crystal structures of the nanoparticles undergo significant changes, resulting in unique nano-effects, for example, surface effects, small-size effects, quantum effects, and macro–quantum tunneling effects. Therefore, nanomaterials have attracted the attention of many researchers as a new high-tech material with a great potential for application. Owing to the unique physical and chemical properties of nanomaterials, many properties of concrete, such as its workability, strength, and durability, can be improved by introducing nanomaterials. Several researchers have conducted numerous investigations based on introducing low-dose nanomaterials for replacing the binding material in concrete mixtures to enhance the qualities of concrete, and have achieved important results [7,8,9]. For instance, the incorporation of NS, nano-CaCO_3_, or nano-TiO_2_ particles has been shown to significantly improve the mechanical properties and endurance of concrete, chiefly owing to their nucleation and micro-aggregate filling effects [10,11,12,13]. In addition to nano-oxides, some nanocarbon materials, comprising of carbon nanotubes, carbon nanofibers, and graphene, have also been shown to dramatically increase the strength, fracture toughness, energy absorption capability, and ductility of ordinary concrete [14,15,16].

Compared with other nanoparticles, NS particles have a higher activity and a particularly large specific surface area; therefore, NS can exhibit a higher level of pozzolanic activity in concrete. In addition, NS can provide nucleation sites for calcium silicate hydrate (C–S–H), and act as a catalyst for pozzolanic reactions to promote the dissolution of tricalcium silicate and the formation of C–S–H [17,18]. Therefore, among these nanomaterial particles, NS particles have received the most attention. Said and Zeidan et al. observed that the compressive strength of concrete blended with 6% of the cement weight of NS was 36% higher than the control concrete [19]. Nazari and Riahi also got an analogous conclusion, but they suggested that NS particles as a fractional replacement of cement at, at most, 4 wt% could improve the formation of C–S–H gel and refine the pore structure of the concrete, and when NS content was in excess of 4 wt%, the strength of the concrete would decrease because of the uneven dispersion of nanoparticles [20]. This is because Said and Zeidan et al. used colloidal NS in their experimental research, and colloidal NS can achieve better dispersion in a cement matrix than powdered NS. Zhang and Ling et al. found that adding 0–1.5% nanosilica particles was optimal for enhancing the compressive strength, flexural strength, and fracture energies of reinforced concrete with polyvinyl alcohol fibers. Moreover, 2.5% nanosilica particles was found to be the best for tensile strength; nevertheless, when the content was greater than 2%, the NS was prone to self-desiccation and flocking together, leading to micro cracks and strength losses in the composite [21,22]. According to the experimental results from most researchers on concrete with the addition of NS, it can be concluded that low doses of NS can significantly enhance the mechanical properties of concrete.

Reducing carbon emissions and promoting carbon neutrality are the main measures taken by authorities around the world to respond to environmental changes [23]. Cement production is the main source of carbon dioxide; therefore, using supplementary cementitious materials (SCMs) such as granulated blast furnace slag and coal fly ash instead of the part of cement to make concrete is an effective way to reduce carbon emissions [24,25]. Coal fly ash is a type of ultrafine particle produced by coal carbon combustion in power plants. As an industrial byproduct, coal fly ash has been broadly utilized for concrete in recent decades, and its recovery has brought significant economic and environmental benefits [26,27,28]. When coal fly ash is used as an SCM to prepare concrete, the loss on ignition is a problem that must be considered. Coal fly ash with a high unburned carbon content increases the conductivity of the concrete, changing the color of the mortar and concrete to black [29]. In addition, coal fly ash with a high carbon content increases the corrosiveness of the metallic parts in concrete [30]. Finally, it can lead to undesirable air entrainment behavior and mixture segregation [31,32]. The durability of high-performance road concrete incorporating coal fly ash was investigated by Li et al., who found that the use of coal fly ash as a cement replacement could significantly improve the permeability resistance and freezing–thawing resistance of concrete [33]. Furthermore, Miguel et al. found that coal fly ash and silica fume improved the durability of mortar and concrete dramatically by effectively inhibiting the alkali–silica reaction [34]. Zhang et al. investigated the effects of silica fume on the compressive strength and fracture properties of coal fly ash concrete. They detected that the compressive strength of concrete containing coal fly ash increased with the increase of silica fume content, but its ability to resist crack propagation gradually decreased [35]. However, an increase in the amount of coal fly ash is not necessarily better. Miguel et al. found that excessive coal fly ash and granulated blast furnace slag will also reduce the carbonation resistance of concrete, especially in badly cured concrete [36,37].

A growing number of concrete structures are of the types frequently affected by impact loads under service conditions. For example, airport runways are subject to impacts from aircraft landing, and offshore structures are subject to the impacts of waves. Therefore, under these impact loads, the structural safety of traditional Portland cement concrete with brittle fracture characteristics poses a significant challenge. Researchers have conducted numerous studies on enhancing the impact resistance of ordinary concrete. For example, Siddique et al. evaluated a concrete that replaced 40% natural sand with fine ceramic aggregate. They observed that the impact energy absorbed by the specimens increased from 0.94 J with plain concrete to 0.99 J [38]. Li et al. used a Hopkinson pressure bar to study the impact resistance of self-compacting concrete with asphalt-coated coarse aggregate. The results showed that applying asphalt onto the surface of coarse aggregate distinctly increased the impact toughness index of the self-compacting concrete, and that when the asphalt layer thickness was 120 µm, the impact resistance was optimal [39]. Gonen found that when the usage rate of waste crumb rubbers in ordinary concrete was 4%, the impact resistance of the specimen was 200% higher than that of a control concrete [40]. Carmichael et al. experimentally studied the influence of 10–50% nanomaterial particle replacement on the impact resistance of concrete relative to a control concrete (without nanomaterials), and found that the nanomaterial concrete had better impact resistance [41]. In addition, incorporating various fiber materials into concrete is considered to be an ideal optimization method for improving the impact resistance of concrete. Numerous research results on fibers have shown that the impact resistance of fiber-containing concrete is significantly improved compared with ordinary concrete, since fiber-containing concrete can consume more impact energy [42,43,44]. Therefore, most of the current methods used to improve the impact resistance of concrete incorporate the addition of energy-absorbing components into the concrete.

It is well known that durability is a critical factor in evaluating the lifespan of concrete. Resistance to chloride permeability and freezing–thawing are two vital elements of this durability. In an environment with a high concentration of chloride ions, for instance, coastal areas and cold areas where deicing salt is used, chloride ions can access the concrete through diffusion, adsorption, and capillarization, causing damage to the internal reinforcement and corrosion of the concrete [45]. Freezing–thawing cycles also pose a severe threat to the durability of concrete structures. In concrete structures, repeated freezing–thawing can cause surface erosion and crack propagation, accelerate the oxidation and corrosion of steel bars, and significantly reduce the service life of the reinforced concrete structure [46,47]. The durability of concrete mixed with NS particles was studied by Du et al.; even at a dosage of 0.3%, the water resistance and chloride ion permeability were greatly improved [8]. They observed that the internal structure of the concrete was denser, specifically in the interfacial transition zone (ITZ), owing to the pozzolanic reaction and nano-filling effect of the NS. Behfarnia and Salemi found that the freezing–thawing resistance of concrete mixes could be noticeably improved by adding nano-Al_2_O_3_ and NS, and that the performance of a concrete containing nano-Al_2_O_3_ was better than that of a concrete containing the same dose of NS [48]. They also found that the appearances of the concrete specimens containing NS after freezing–thawing cycles were distinctly improved, and that the quality loss was significantly reduced. According to Zhang et al., the amount of NS added to a mixture significantly improves the freezing and thawing resistance of a high-performance concrete [49]. Gonzalez et al. observed that adding NS to a concrete mixture can reduce the external damage caused by freezing–thawing cycles, owing to the production of denser and lower-permeability concrete [50].

Although there have been a large number of studies on the impact resistance of ordinary concrete, and the mechanical properties and durability of concretes containing NS, results correlated with the impacts of NS particles on the impact resistance and durability of coal fly ash concretes are relatively rare. Therefore, in this study, the effects of different NS dosages on the impact resistance, chloride penetration resistance, and freezing–thawing resistance of concretes containing coal fly ash were studied using a drop-weight impact test, single-sided freezing–thawing test, and rapid chloride migration (RCM) test, respectively; then, the optimal NS content was determined.

## 2. Experimental Program

### 2.1. Materials

In this study, P·I 42.5 Portland cement produced by Shandong Luneng Cement Co., Ltd. (Laiwu, China) and first-grade coal fly ash from Datang Luoyang Thermal Power Plant (Luoyang, China) that met Chinese standards were used [51,52]. The properties and composition of the cement and coal fly ash are listed in Table 1, Table 2 and Table 3, respectively. A laser diffraction-type particle size analyzer was used to measure particle size distribution. The coarse aggregate was composed of continuously graded gravel with particle sizes of 4.75 mm to 26.5 mm; the fine aggregate was well-graded natural river sand. The properties of the natural river sand and coarse aggregate are listed in Table 4 and Table 5, respectively. A polycarboxylate superplasticizer was used to facilitate the diffusion of the nanoparticles. The NS was produced by Zhejiang Wanjing New Material Co., Ltd. (Hangzhou, China), and was included by partially replacing the cement by weight in the concrete mix. NS purity was 99.6%, and the main properties of the NS are shown in Table 6. The appearance of the powdery NS is exhibited in Figure 1.

A laser diffraction-type particle size analyzer (Microtrak-9320HRA, Aichi-ken City, and Japan.) was used to measure particle size distribution (PSD), the PSD of coal fly ash and cement is shown in Figure 2, and a scanning electron microscope image of NS particles is shown in Figure 3. Among them, coal fly ash particles with a particle size of less than 15 µm accounted for 80% of the total volume of the coal fly ash, and 80% of the cement particles had a particle size of less than 25 µm. In addition, it can be seen from Figure 3 that the average particle size of the NS particles was about 30 nm.

### 2.2. Mix Proportions and Specimen Preparation

The experiment in this study was based on the concrete mix design specification JGJ 55-2011 [53], and the concrete mix was designed for C45. According to the related research on coal fly ash concrete, 15% of the weight of the cementitious material was replaced by coal fly ash to ensure the performance of the concrete [54]. While keeping the water–cement ratio (0.37) and sand ratio (646 kg/m^3^) constant for all of the mixtures, six different mix proportions were prepared, including a control group and five groups containing different NS dosage mixtures. Among them, NS was used to replace cement in proportions of 1%, 2%, 3%, 4%, and 5%, respectively. The concrete mix proportions are presented in Table 7 where PC denotes pure concrete, and NSC1, NSC2, NSC3, NSC4, and NSC5 denote NS contents in the concrete at 1%, 2%, 3%, 4%, and 5% of the cement weight, respectively. To maintain a constant slump level, different dosages of superplasticizer were used for the five groups of mixtures.

The uniform dispersion of the NS is the key to ensuring the quality of a concrete, and is also the basis for achieving excellent characteristics in a specimen [55]. To achieve better dispersion of the SiO_2_ nanoparticles and fabricate homogenous and uniform concrete mixtures, the following procedures were employed. 

The NS and superplasticizer are added into water and agitated evenly for 90 s.The coarse and fine aggregates were added into the wetted concrete mixer and mixed for 90 s.The cement and coal fly ash were added into the mixer, and the ingredients were mixed for 90 s.Mixing water (containing NS and superplasticizer) was added into the mixer and mixed for 90 s.The remaining water or the entire quantity of water (in the control mixtures) was added into the mixer and mixed for approximately 90 s.

Immediately after mixing, the fresh mixture was cast into an oiled cube mold with a size of 150 × 150 × 150 mm^3^. The specimens were demolded after being placed horizontally at room temperature for 24 h, and were then placed in water at room temperature and removed for testing after 28 days of curing.

### 2.3. Experimental Methodology

#### 2.3.1. Mechanical Properties Tests 

The three tests (compressive strength, flexural strength, and splitting tensile strength) were all in accordance with JTG E30-2005 [56]. Cube specimens with a size of 150 × 150 × 150 mm^3^ were used in the compressive and splitting tensile strength tests. Specimens with sizes of 100 × 100 × 400 mm^3^ and a flexure testing machine with a maximum range of 1000 kN (meeting the performance test requirements) were used in the flexure strength test. 

#### 2.3.2. Impact Resistance Test

The impact tests were performed on a drop hammer test machine, with specimen sizes of 150 × 150 × 150 mm^3^. The impact energy range of the falling weight was 50–2000 J. An impact process was defined as a cycle after each cycle was completed, and the surface of the specimen was carefully observed. When the first crack appeared in the specimen, the number of impacts at this time was denoted as “N1” for the initial cracks in the concrete specimen. The next impact cycle was continued until the width of a crack on the surface of the specimen expanded to 1 mm, when the test was terminated. At this time, the number of impacts was denoted as “N2.” The impact test results are shown in Figure 4a.

#### 2.3.3. Resistance to Chloride Penetration Test

The rapid chloride migration (RCM) method proposed by Tang and Nilsson [57] was used in the chloride migration coefficient test (CDCT), and the samples used for the CDCT were cylinders. The CDCT is shown in Figure 4b. At the end of the test, the samples were divided into two parts along the diameter, and silver nitrate solution (0.1) M was quickly sprayed on the split face. After standing for approximately 15 min, the color development was observed, and the penetration depth curve was drawn using a pen. The samples were divided into ten equal parts along the sample diameter and the penetration depth of the chloride was measured. After the test was completed, the diffusion coefficient of the chloride ions was determined using the following formula [58], as follows:(1)$$D_{RCM}=\frac{0.0239\times \left(273+T\right)L}{\left(U-2\right)t}\left(X_{d}-0.0238\sqrt{\frac{\left(273+T\right)LX_{d}}{U-2}}\right)$$
where $D_{RCM}$ is the non-steady-state migration coefficient of the concrete, accurate to 0.1 × 10^−12^ m^2^/s; U is the test load voltage; T is the average value of the initial and final temperature of the anode solution; L is the specimen thickness; $X_{d}$ is the average value of the chloride ion penetration depth; and t is the electricity test time.

#### 2.3.4. Freezing-Thawing Resistance Test

To accurately simulate an environment in which concrete undergoes salt freezing in the atmosphere, a single-sided freezing–thawing test method closer to the actual freezing–thawing damage environment of concrete was adopted in this study, as shown in Figure 4c. The freeze–thaw resistance of the concrete was evaluated by measuring the RDEM of the concrete after 4, 16, and 28 freeze–thaw cycles. To obtain accurate test data, the average of each group of five test pieces was taken as the relative elastic modulus test value.

## 3. Results and Discussion

### 3.1. Effect of NS on Mechanical Properties of Concrete

Compared with the control concrete, the mechanical properties of the coal fly ash concretes mixed with NS were significantly improved. To make the results more illustrative, the effects of different NS contents on the 28-day compressive and flexural strength of the specimens and contents of NS are shown in Figure 5 and Figure 6, respectively. With the increase of NS content, the compressive strength of the concrete presented a changing law that first heightened and then dropped; the compressive strength reached a maximum value of 52.3 MPa when the content was 3%, which was an increment of 15.5% relative to the control concrete. Analogously, the concrete with 3% NS content had the maximum flexural strength, reaching 8.86 MPa. This was 27.3% higher than that of the control concrete, and was similar to the conclusion drawn by Rong et al. [59]. Notably, even though the flexural and compressive strengths of the concrete with 5% NS content were not the highest, they were greater than those of ordinary concrete without NS. The effects of the NS content changes on the splitting tensile strength of the concrete are shown in Figure 7. When the NS content of NS was 2%, the splitting tensile strength of the concrete reached the maximum 4.51 of MPa, an increase of 19% relative to the control concrete, consistent with the results of P. Ganesh et al. [60,61].

The pozzolanic reaction and nano-filling effect are the two main reasons for the enhanced strength of the concrete [62]. In particular, owing to the large specific surface areas of NS particles, which have ultra-high reactivity, NS particles can interact with water molecules in the concrete mixture and generate silanol groups (Si–OH). Then, Si–OH reacts with the Ca^2+^ in the calcium hydroxide (Ca(OH)_2_) crystals and forms a C–S–H gel [63]. Moreover, the unreacted NS particles are dispersed in smaller spaces and fill the void, refining the pore structure, and improving the compactness of the concrete. As previously mentioned, NS particles have extremely large specific surface areas, and when the NS added exceeds the optimal dosage, agglomeration phenomena can occur in the mixture [64]. Simultaneously, the nanoparticles have strong water absorption; accordingly, excessive NS absorbs the water originally required for the cement hydration, resulting in insufficient cement hydration and a decreased strength of the concrete [65].

### 3.2. Impact Resistance

The influences of the NS content on the number of blows at the first crack and at the failure of the concrete are shown in Figure 8. From Figure 8, when the dosage of NS was less than 2%, the number of hammerings increased with the increase of NS substitution rate. When the percentage replacement of NS was 2%, the number of hammerings at the first crack and at failure reached maximums of 37 and 40, respectively, which were 23.3% and 29% higher than those of the control concrete, respectively. When the percentage replacement of NS was greater than 2%, the initial and final crack blow counts of concrete began to decline; when the substitution rate of NS was 5%, the number of blows at the first crack and at the failure of the concrete were both reduced to 27, i.e., reductions by 10% and 12.9% relative to the control concrete, respectively. This showed that excessive NS can not only reduce the improvement effect on the impact resistance of the concrete, but can even reduce the impact resistance.

The effect of the NS content on the impact energy of the concrete is shown in Figure 9. It can be seen that as the content of NS changed, the impact energy difference of the concrete varied from 0 to 150 J, and the maximum impact energy difference was only 150 J. This indicates that the NS can improve the initial crack impact resistance of the concrete, but after the initial crack, the concrete continues to store energy. The capacity was not significantly improved, and the initial cracking to failure was relatively rapid, without changing the brittleness of the concrete. The impact failure mode of the specimens also confirmed this conclusion. A comparison of the impact damage morphologies of ordinary concrete specimens and concrete specimens containing NS is shown in Figure 10. The concrete specimens containing NS also fractured in half along the impact direction after being impacted, indicating that the addition of NS did not ameliorate the impact damage morphology of the concrete specimens, and that concrete specimens mixed with NS still show brittle failures similar to that of ordinary concrete.

Owing to its high pozzolanic activity, NS can react rapidly with the Ca(OH)_2_ crystals arranged in the ITZ between the cement paste and aggregate to form a C–S–H gel [66]. Furthermore, the NS particles have a filling effect, making the microstructure denser and uniform, and thus improving the strength of the concrete [62]. Moreover, Carmichael et al. found that there is a strong positive correlation between the impact strength and compressive strength of NS concrete [41]. The increase in the number of initial crack and final crack hammerings of the concrete specimens containing NS may be owing to the addition of the NS, which improves the structure and strength of the concrete mixture. However, when excessive nanoparticles are added to the mixture, they cannot be uniformly dispersed in the cement paste; therefore, agglomeration areas will appear in the cement paste, and these areas will become weak zones [67]. The generation of these weak zones adversely affect the internal structure of the concrete, further reducing the impact resistance of the concrete specimen.

### 3.3. Chloride Penetration Resistance

Figure 11 clearly shows that the chloride penetration resistance of the concrete gradually enhances with the increase of NS particle content. As illustrated in Figure 11, the chloride diffusion coefficient of the concrete first decreases and then increases with the increase in the percentage replacement of NS. When the replacement level of NS was 2%, the chloride ion diffusion coefficient reached the minimum, and the concrete had the best anti-chloride ion penetration performance. The chloride ion diffusion coefficient of concrete with a 40% replacement level of NS was 68.3% lower than that of the control concrete. When the replacement level of NS increased to 3%, 4%, and 5%, respectively, the chloride ion diffusion coefficient of concrete gradually increased, but was still decreased by 57.3%, 51.2%, and 20.7%, respectively, compared with that of the control concrete. Similar conclusions can be found in the literature [62,68].

The chloride penetration resistance of concrete is mostly dependent on the pore sizes of the concrete. Metha et al. stated that pores in concrete with pore sizes larger than 100 nm will significantly reduce the mechanical properties and permeability resistance of the concrete [69]. NS is added to concrete as a nanoscale material with an average particle size smaller than that of other admixtures in concrete, and can therefore transform harmful pores into harmless ones, and reduce the porosity of the concrete. Correspondingly, it can also inhibit the generation of concrete pores, refine the pore size (specifically at the ITZ), and make the microstructures of the concrete denser and more homogenous. In addition, the NS in the cement matrix can effectively block or cut off capillaries in the concrete, forming tortuosity and more disconnected transport channels, further improving the chloride permeability resistance of the concrete sample [8]. As mentioned earlier, owing to the large specific surface areas of the nanoparticles, the NS particles will agglomerate and become unable to disperse uniformly in the cement matrix after being added to the mixture at higher replacement ratios of NS, and the chloride resistance permeability will be lower than that of the concrete with the optimum NS threshold. 

### 3.4. Freezing-Thawing Resistance

The effects of different amounts of NS on the RDEM of the concrete after 0, 4, 16, and 28 freezing–thawing cycles are shown in Figure 12. For specimens with the same mix proportions, the RDEM of the concrete decreased gradually with an increase in the number of freezing–thawing cycles, and it can be seen from the slope of the curve that the decreasing amplitude gradually increased. With an NS content of 5% in the concrete specimens, for example, the RDEM decreased by less than 10% after four cycles. When the freezing–thawing cycles were repeated 28 times, the RDEM decreased to less than 60%, while the RDEM of concrete decreased sharply with the increase of freeze–thaw cycles, rather than showing a linear decrease. For specimens with different mix proportions, when NS content increased from 0 wt% to 5 wt%, after the same number of freeze-thaw cycles, the RDEM of the concrete first increased and then decreased, and the freezing-thawing resistance of the concrete first improved and then decreased. When the NS content exceeded 2%, the RDEM exhibited a downward trend. In particular, when the content of NS was 5%, the RDEM of the concrete after the freeze–thaw cycles was lower than that of the concrete without NS, indicating that excessive NS will reduce the freezing–thawing resistance of the concrete. 

The internal cause of the damage to concrete caused by freezing–thawing cycles is mainly determined by the porosity and pore distribution of the concrete [70], and the mechanisms of concrete salt-frost damage are very complicated. Compared with pure frost damage without deicing salt, the freezing damage caused by salt not only includes physical damage, but also chemical damage. The physical damage is revealed as the presence of salt increases the water retention capacity of the concrete, increasing the internal osmotic pressure of the concrete, and aggravating the damage to the concrete. The chemical damage refers to the fact that deicing salt can react with active aggregates to form alkali aggregates, while calcium chloride and Ca(OH)_2_ react to form compound salts. This destroys the state of the C–S–H gel, promotes the scaling of the concrete surface, and destroys the freezing–thawing resistance of the concrete [71].

In the early stages, i.e., after experiencing only a few freezing–thawing cycles, the interiors of the concrete specimens had not yet undergone freezing–thawing cracking, and the expansion of the micro-cracks was small. In addition, the generation of frozen water in the early stages would squeeze the concrete, increasing the density and improving the RDEM of the concrete. The experimental data in this study also show that the RDEM of concrete was above 90% after four freeze-thaw cycles. As the number of freeze–thaw cycles increased, the pressure of the water inside the pores owing to icing and expansion would damage the microstructure inside the concrete [72,73], and the capillaries would develop into microcracks. When the microcracks developed into macrocracks, they caused serious damage to the concrete. When a small dosage of NS was incorporated, the nanoparticles could be evenly dispersed in the cement matrix, and the NS particles acted as nanofillers to refine the internal pores of the concrete and reduce the porosity of the concrete. Simultaneously, the NS blocked and isolated the capillary channels inside the concrete and inhibited the invasion of the sodium chloride solution. As a pozzolanic material, NS reacts with the Ca(OH)_2_ formed by the calcium silicate hydration to form a more orderly C–S–H gel [74], making the concrete mixture more homogeneous and compact. When a large dosage of NS is incorporated, the nanoparticles cannot fully participate in hydration, owing to agglomeration of NS or insufficient Ca(OH)_2_ for reaction. As a result, the porosity of the concrete increases, and the freezing–thawing resistance of the concrete decreases.

## 4. Conclusions

In this study, the influences of NS particles on the mechanical properties, impact resistance, and durability resistance of coal fly ash concretes were analyzed and discussed, based on the experimental results obtained through mechanical properties tests, impact resistance tests, resistance to chloride penetration tests, and freezing-thawing cycle resistance tests. The following conclusions were drawn:(1)Coal fly ash concrete containing NS significantly improved the mechanical properties of concrete specimens relative to a control concrete (without NS), which was due to the reaction of NS particles with Ca(OH)_2_ at the ITZ to produce more C-S-H gel and the densification of the microstructure. However, the optimum amount for each mechanical property was different; the splitting tensile strength reached a maximum when the NS replacement level was 2%, and when the NS replacement level was 3%, the compressive and flexural strengths reached their maximums.(2)Adding NS particles can dramatically improve the impact resistance at the first cracking of the concrete; however, the speed from initial cracking to failure was faster, and the brittleness of the concrete did not change. After being impacted, the concrete specimens containing NS were divided into two halves along the impact direction, and the concrete specimens still exhibited brittle failures, as in ordinary concrete.(3)Compared with the control concrete, the low dosage of NS can greatly ameliorate the chloride penetration resistance and freezing–thawing resistance of the concrete. At the 2% replacement level, a uniform dispersion of NS particles was easily achieved. The improvement of the pore structures in the concrete and filling effect of the NS were responsible for the reduction in the chloride diffusion coefficient, and ameliorated the freezing–thawing resistance of the concrete.

## Figures and Tables

**Figure 1 nanomaterials-11-01296-f001:**
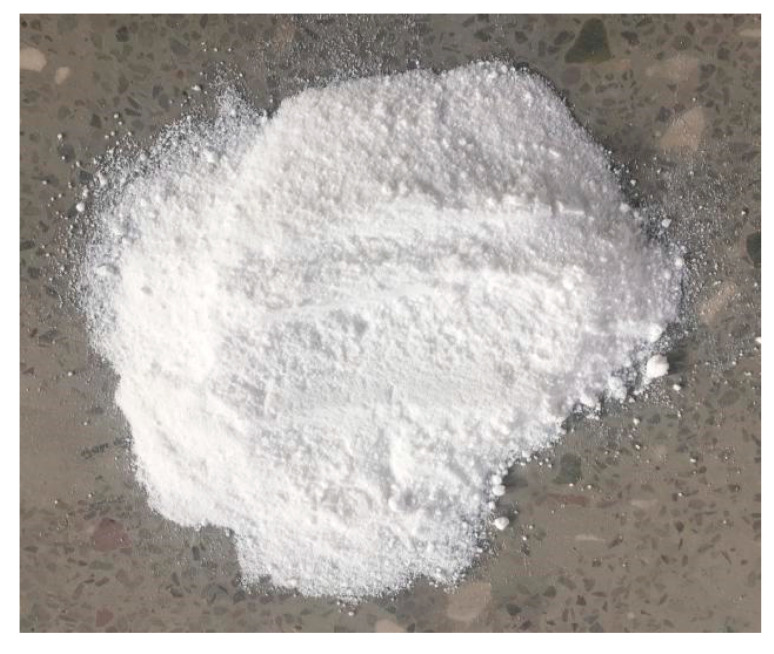
Appearance of powdery nanosilica (NS).

**Figure 2 nanomaterials-11-01296-f002:**
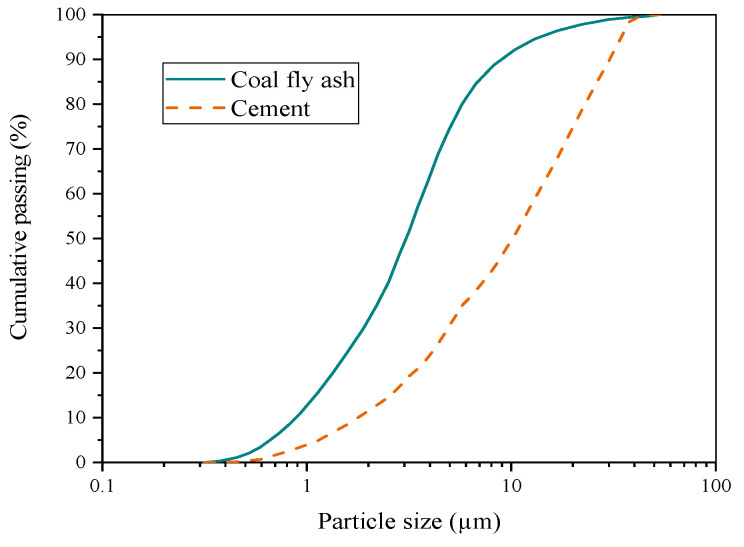
PSD of coal fly ash and cement.

**Figure 3 nanomaterials-11-01296-f003:**
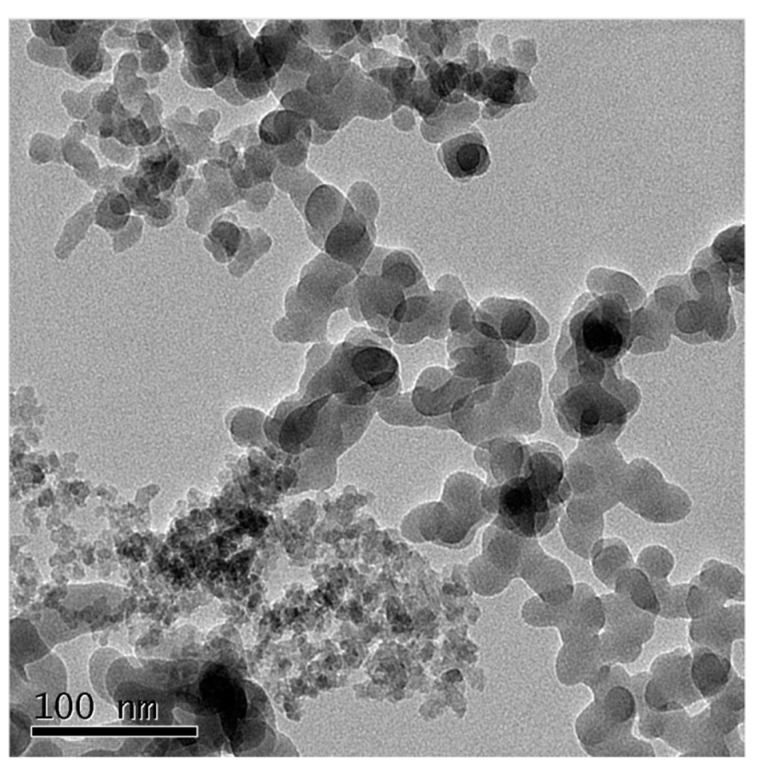
Scanning electron microscope (SEM) image of NS particles.

**Figure 4 nanomaterials-11-01296-f004:**
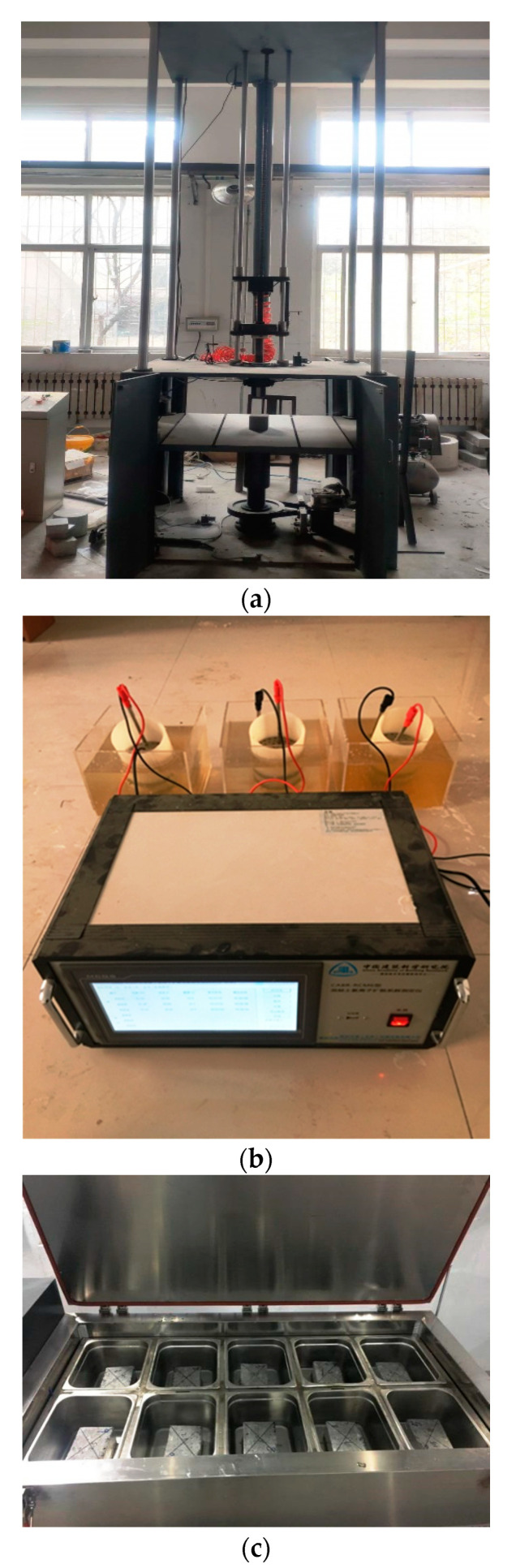
Test apparatus, (**a**) impact testing machine, (**b**) Test device for rapid chloride migration (RCM) method, (**c**) Single-sided freezing–thawing test.

**Figure 5 nanomaterials-11-01296-f005:**
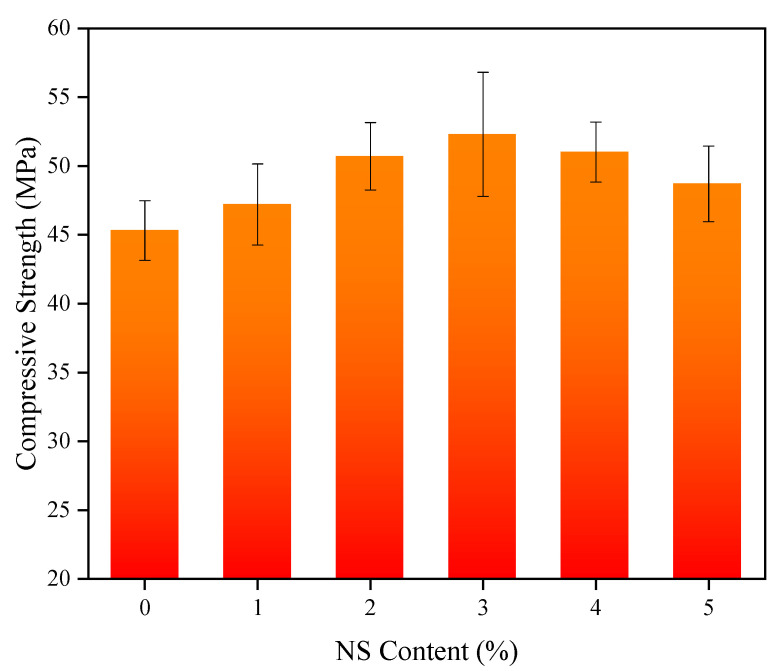
Compressive strength of concrete with NS.

**Figure 6 nanomaterials-11-01296-f006:**
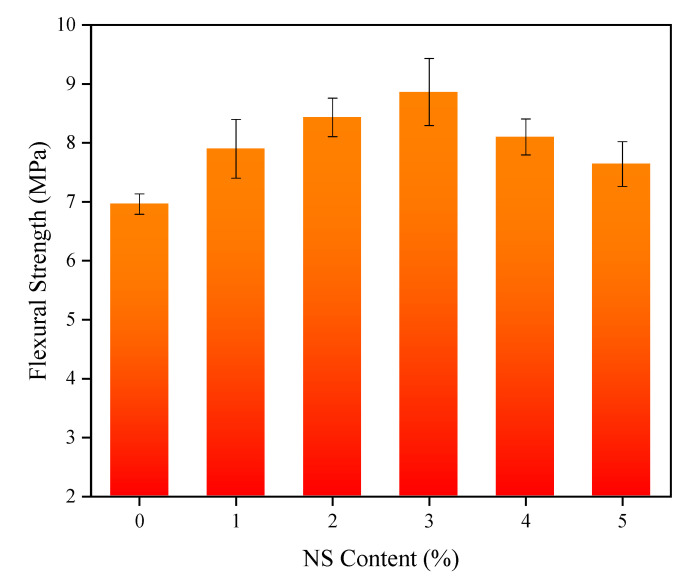
Flexural strength of concrete with NS.

**Figure 7 nanomaterials-11-01296-f007:**
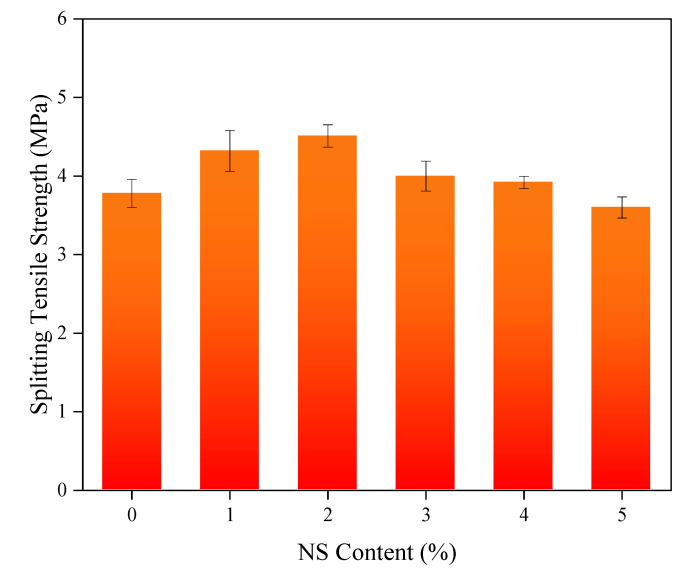
Splitting tensile strength of concrete with NS.

**Figure 8 nanomaterials-11-01296-f008:**
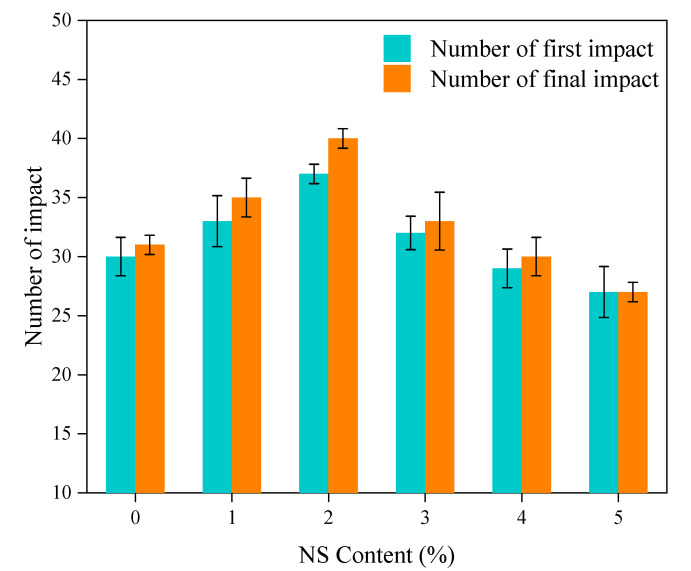
Influence of the content of NS on the number of impacts.

**Figure 9 nanomaterials-11-01296-f009:**
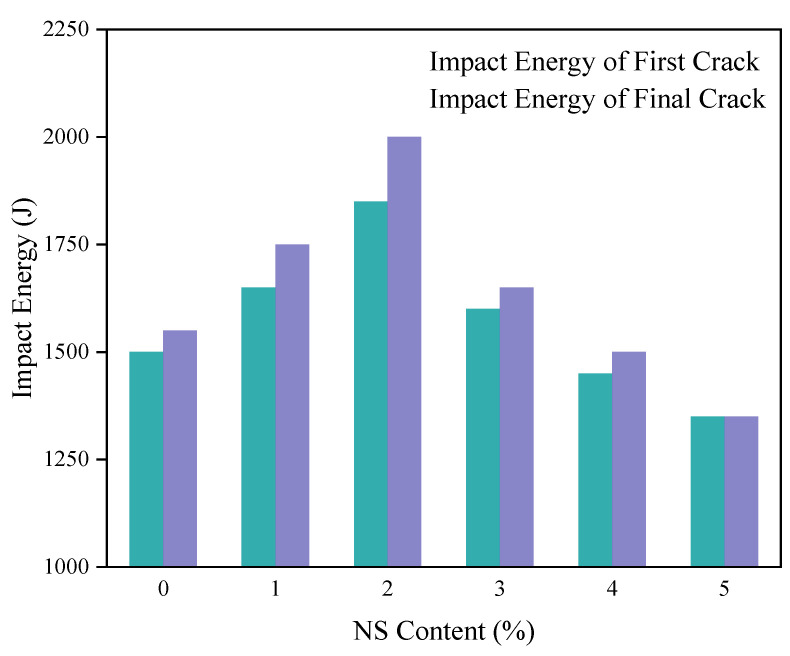
Impact energy of concrete with NS.

**Figure 10 nanomaterials-11-01296-f010:**
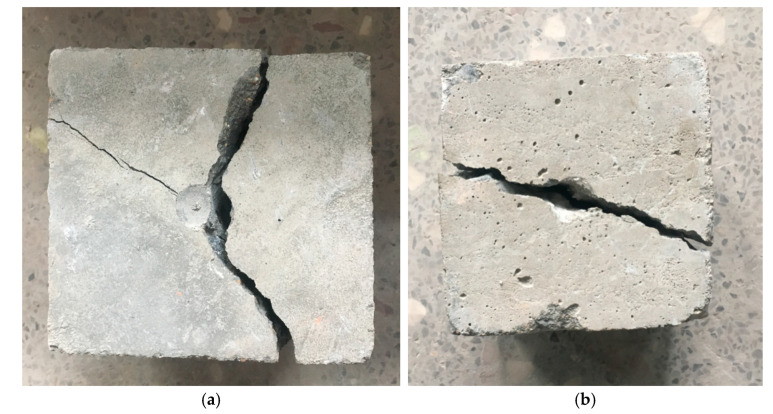
Impact damage diagram of concrete containing NS; (**a**) concrete without NS; (**b**) concrete mixed with 2% NS.

**Figure 11 nanomaterials-11-01296-f011:**
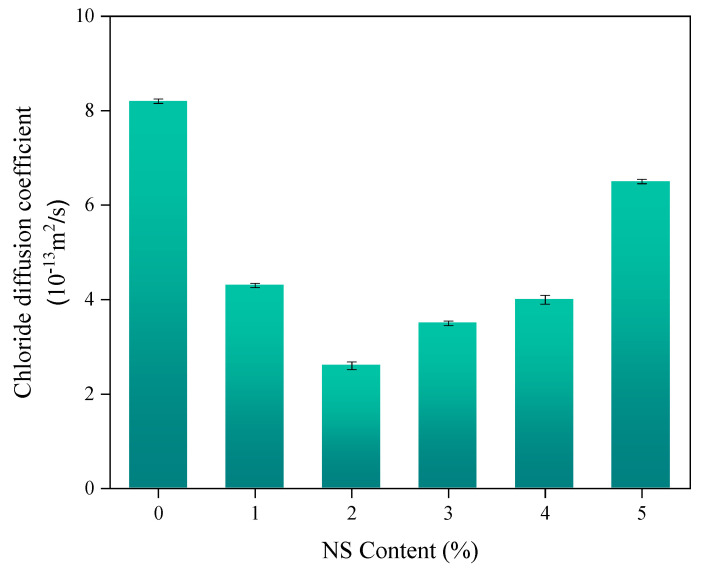
The influence of NS content on the chloride diffusion coefficient of concrete.

**Figure 12 nanomaterials-11-01296-f012:**
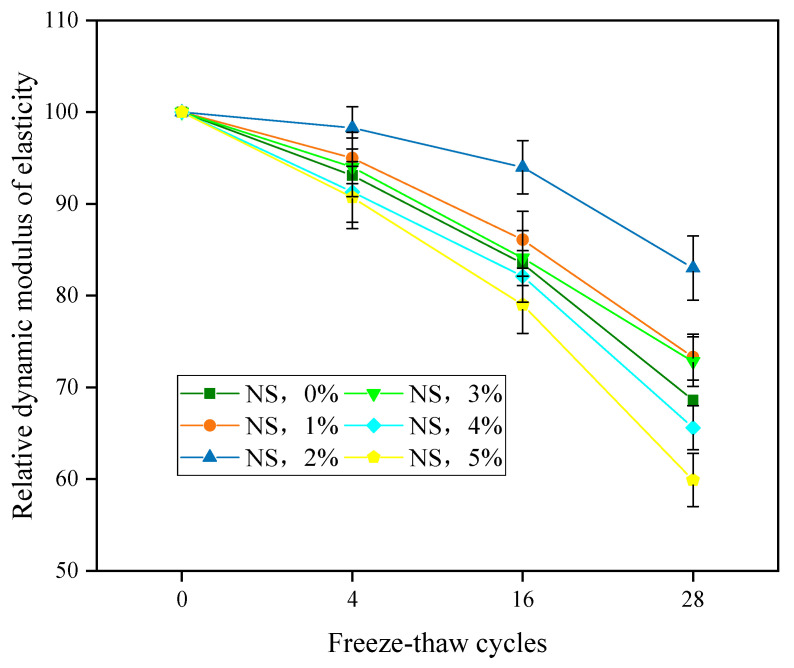
RDEM of concrete specimens incorporated with NS.

**Table 1 nanomaterials-11-01296-t001:** Properties of cement.

Density (g/cm^3^)	Setting Time (min)		Compressive Strength (MPa)		Flexural Strength (MPa)	
	Initial Setting	Final Setting	3d	28d	3d	28d
3.26	85	285	28.1	54.6	5.96	9.45

**Table 2 nanomaterials-11-01296-t002:** Properties of coal fly ash.

Degree of Fineness (%)	Water Demand Ratio (%)	Water Content (%)	Ignition Loss (%)	Sulfur Trioxide Content (%)	Free Calcium (%)
9.21	91.1	0.5	5.24	1.21	0.19

**Table 3 nanomaterials-11-01296-t003:** Composition of cement and coal fly ash.

Chemical Composition (%)	Portland Cement	Coal Fly Ash
SiO_2_	21.05	52.12
Al_2_O_3_	5.28	17.86
Fe_2_O_3_	2.57	6.57
CaO	63.14	9.12
MgO	3.58	3.26
Na_2_O	0.17	2.38
K_2_O	0.58	2.05
SO_3_	2.39	0.23

**Table 4 nanomaterials-11-01296-t004:** Properties of natural river sand.

Fineness Modulus	Mica (%)	Silt Content (%)	Robustness (%)	Sulfide (%)	Apparent Density (kg/m^3^)	Bulk Density (kg/m^3^)
2.7	0.2	1.5	5.0	0.3	2560	1540

**Table 5 nanomaterials-11-01296-t005:** Properties of coarse aggregate.

Moisture Content	Silt Content (%)	Robustness (%)	Crushing Value	Apparent Density (kg/m^3^)	Bulk Density (kg/m^3^)
0.36	1.2	5.0	7.0	2735	1401

**Table 6 nanomaterials-11-01296-t006:** Properties of NS.

Content (%)	pH	Average Particle Size (nm)	Specific Surface Area (m^2^/g)	Loss on Drying (%)	Ignition Loss (%)	Apparent Density (g/L)
99.6	6.2	30	200	1.0	1.0	56

**Table 7 nanomaterials-11-01296-t007:** Concrete mix proportions.

Mixture ID	Water (kg/m^3^)	Cement (kg/m^3^)	Coal Fly Ash Replacement Ratio (%)	Fly Ash (kg/m^3^)	Sand (kg/m^3^)	Coarse Aggregate (kg/m^3^)	NS (%)	Superplasticizer (%)
PC	190	437	15	77	646	990	0	0
NSC1	190	432.63	15	77	646	990	1	0.2
NSC2	190	428.26	15	77	646	990	2	0.4
NSC3	190	423.89	15	77	646	990	3	0.6
NSC4	190	419.52	15	77	646	990	4	0.8
NSC5	190	415.15	15	77	646	990	5	1.0

## Data Availability

The data presented in this study are available on request from the corresponding author.

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
