# Peer review of "Effect of Nano Silica Particles on Impact Resistance and Durability of Concrete Containing Coal Fly Ash"

_nanomaterials, 2021, doi:10.3390/nano11051296_

Round 1
Reviewer 1 Report
Manuscript ID: nanomaterials-1181104-peer-review-v1
Nanomaterials
Effect of nano silica particles on impact resistance and durability of concrete containing fly ash
Reviewer comments:
SUMMARY
The manuscript deals with a good investigation on the use of nano-silica (NS) particles in concrete made with coal fly ash. Mechanical strength, impact resistance, chloride penetration resistance, and freezing-thawing resistance were assessed. This is a topic that has not been widely covered in the literature, therefore, this a subject of great interest, but it is somehow limited in the analysis and application of these results.
MAIN IMPRESSIONS
This paper has an undeniable practical usefulness. However, from a scientific point of view, the following issues must be addressed: i) Particle size distribution (PSD) of the cement, coal fly ash and NS should be included in the investigation; and ii) a deeper discussion and comparison with other studies is necessary.
MORE DETAILED COMMENTS
Lines 1 & 82 & so on: “Coal fly ash” could be more precise than “fly ash”.
Lines 37-39: “…concrete applications becomes wider, higher requirements for concrete…”. Furthermore, I suggest introducing the topic of the future for the manufacture of cements and concretes, in order to reduce the carbon impact of ordinary Portland cement and improve the circular economy. Then, I suggest mentioning the role of the Roadmaps of the Cement Industry. Currently, some Roadmaps of the Cement Industry consider several levers to achieve the carbon neutrality. For instance, Energies 2020, 13, 3452. https://doi.org/10.3390/en13133452. Could you please add some references? The use of Supplementary cementitious materials (SCM) to reduce the OPC content or the use of blended cements is one lever.
Line 54: subscripts in CaCO3 and nano-TiO2.
Line 104: 120 um è µm.
Line 42 & 56: Nested references [1-7] & [11-16]: I suggest explaining the most significant findings for each reference.
Line 158: Table 1: Chemical properties of Portland cement, coal fly ash and nano-silica (NS) particles should be added.
Line 117: I agree that durability is a critical factor in evaluating the lifespan of concrete. However, chloride resistance and freezing-thawing resistance are not the only key factors. It is well-known that coal fly ash and nano-silica (NS) improve both properties, but they decrease carbonation resistance, especially in bad cured concretes. Could you please discuss in the introduction this point? For instance: Carbon Dioxide Absorption by Blast-Furnace Slag Mortars in Function of the Curing Intensity. Energies 2019, 12(12), 2346; https://doi.org/10.3390/en12122346; Effect of curing time on granulated blast-furnace slag cement mortars carbonation. Cement and Concrete Composites 90 (2018) 257–265. https://doi.org/10.1016/j.cemconcomp.2018.04.006
Line 117: Another important aspect with regard to the durability is the Alkali-Silica Reaction (ASR). Could you please add the references on this topic?. For instance: Sustainable and Durable Performance of Pozzolanic Additions to Prevent Alkali-Silica Reaction (ASR) Promoted by Aggregates with Different Reaction Rates. Appl. Sci. 2020, 10, 9042. https://doi.org/10.3390/app10249042
Line 150: Could you please add the reference for the P·I 42.5 Portland cement?
Line 151: Could you please add the supplier of the cement and coal fly ash?
Line 151: Why have you decided to keep constant the amount of coal fly ash (77 kg/m3)?
Line 154: Could you please add the type of liquid superplasticizer?
Line 159: Particle size distribution (PSD) of the cement, coal fly ash and NS should be included in the investigation.
Line 159: A LOI of 5.24 could be considered high. Unburned carbon is an undesirable constituent of fly ashes to be utilized in the reinforced concrete construction. Therefore, it should be mentioned in the introduction. The problem is that the unburned carbon in fly ashes has several detrimental effects on the concrete. Especially, it increases the electrical conductivity of the concrete, changes the color of mortar and concrete (they may appear black), etc. Moreover, the water/(cement+fly ash) ratio, needed to obtain a cement paste with a required rheological properties or consistency, is higher for fly ashes with a high carbon content, increasing the corrosivity of metallic parts incorporated in the concrete. Finally, it causes a poor air entrainment behavior and mixture segregation. The following papers deal with this topic:
- Freeman, E., Gao, Y-M., Hurt, R. and Suuberg, E.: 1997, Interactions of carbon-containing fly ash with commercial air-entraining admixtures for concrete, Fuel, 76, no. 8, 761–765. https://doi.org/10.1016/S0016-2361(96)00193-7
- Ha, T.H., Muralidharan, S., Bae, J.H., Ha, Y.C., Lee, H.G., Park, K.W. and Kim, D.K.: 2005, Effect of unburnt carbon on the corrosion performance of fly ash cement mortar, Construction and Building Materials, 19, 509–515. https://doi.org/10.1016/j.conbuildmat.2005.01.005
- Ehsan Ghafari, Seyedali Ghahari, Dimitri Feys, Kamal Khayat, Aasiyah Baig, Raissa Ferron. Admixture compatibility with natural supplementary cementitious materials, Cement and Concrete Composites, Volume 112, 2020, 103683, https://doi.org/10.1016/j.cemconcomp.2020.103683
- Lim, W. Lee, H. Choo, C. Lee, Utilization of high carbon fly ash and copper slag in electrically conductive controlled low strength material, Construction and Building Materials, Volume 157, 2017, Pages 42-50, https://doi.org/10.1016/j.conbuildmat.2017.09.071
Line 160: “pH” instead of “PH”.
Line 174: I agree that “The uniform dispersion of the NS is the key to ensuring the quality of a concrete”. Could you please add some references? For instance,
Effect of silica fume fineness on the improvement of Portland cement strength performance. CONSTRUCTION AND BUILDING MATERIAL Vol. 96, 2015, pp. 55–64. http://dx.doi.org/10.1016/j.conbuildmat.2015.07.092
Line 199: What do you mean saying “In perchloride environments”. Perchloride is a chloride that contains more chlorine than other chlorides of the same element.
Line 210: “… The RCM method proposed by Tang and Nilsson [49] was used in the chloride dif. …”. Therefore, the Chloride diffusion coefficient is calculated based on the guideline of NT BUILD 492? NORDTEST, “Chloride migration coefficient from non-steady-state migration experiments,” NT BUILD 492, 1999. The scope is ”This procedure is for determination of the chloride migration coefficient in concrete, mortar or cement-based repair materials from non-steady-state migration experiments.”. Then, I suggest to replace ”diffusion” by ” migration”.
Line 236: “3. Results and discussion”: In order to optimize the concrete mixture, you should follow a statistical procedure. For instance, check the following reference and discuss it in the introduction: Granulated Blast-Furnace Slag and Coal Fly Ash Ternary Portland Cements Optimization. Sustainability 2020, 12, 5783. https://doi.org/10.3390/su12145783
Line 307: “… microstructure more denser … “ or “… microstructure denser … “?
Line 314: Agglomeration of NS has been reported in the literature: Combined effect of nano-SiO2 and nano-Fe2O3 on compressive strength, flexural strength, porosity and electrical resistivity in cement mortars. Mater. Construcc. Vol. 68, Issue 329, January–March 2018, e150. https://doi.org/10.3989/mc.2018.10716
Line 393: Calcium hydroxide in solution reacts with NS. Then, why do you say “…crystals for reaction”?.
Line 418: In “ … in the chloride diffusion coefficient, …”, I suggest to replace ”diffusion” by ” migration”.
RECOMMENDATION
In conclusion, Major changes have been proposed.
Author Response
1.This paper has an undeniable practical usefulness. However, from a scientific point of view, the following issues must be addressed: i) Particle size distribution (PSD) of the cement, coal fly ash and NS should be included in the investigation; and ii) a deeper discussion and comparison with other studies is necessary.
Response: Thank you for your suggestion. i) In the revised manuscript, a laser diffraction-type particle size analyzer was used to measure particle size distribution (PSD) of cement and coal fly ash, and a scanning electron microscope (SEM) was used to analyze particle size of NS. ii) We have conducted more in-depth discussions and comparisons with other studies.
2.Lines 1 & 82 & so on: “Coal fly ash” could be more precise than “fly ash”.
Response: In the revised manuscript, we have replaced fly ash with coal fly ash.
3.Lines 37-39: “…concrete applications becomes wider, higher requirements for concrete…”. Furthermore, I suggest introducing the topic of the future for the manufacture of cements and concretes, in order to reduce the carbon impact of ordinary Portland cement and improve the circular economy. Then, I suggest mentioning the role of the Roadmaps of the Cement Industry. Currently, some Roadmaps of the Cement Industry consider several levers to achieve the carbon neutrality. For instance, Energies 2020, 13, 3452. https://doi.org/10.3390/en13133452. Could you please add some references? The use of Supplementary cementitious materials (SCM) to reduce the OPC content or the use of blended cements is one lever.
Response: In the revised manuscript, we have cited the mentioned article, and we have added discussions on the subject of green structures and eco-friendly materials, especially on the solutions to carbon dioxide emissions and the greenhouse effect.
4.Line 54: subscripts in CaCO3 and nano-TiO2.
Response: We have corrected this mistake in the revised manuscript.
5.Line 104: 120 um è µm.
Response: We have corrected this mistake in the revised manuscript.
6.Line 42 & 56: Nested references [1-7] & [11-16]: I suggest explaining the most significant findings for each reference.
Response: The significant findings of these references were discussed in more detail in the revised manuscript.
7.Line 158: Table 1: Chemical properties of Portland cement, coal fly ash and nano-silica (NS) particles should be added.
Response: We have added the chemical properties of Portland cement, coal fly ash in Table 3, and the purity of silica was added.
8.Line 117: I agree that durability is a critical factor in evaluating the lifespan of concrete. However, chloride resistance and freezing-thawing resistance are not the only key factors. It is well-known that coal fly ash and nano-silica (NS) improve both properties, but they decrease carbonation resistance, especially in bad cured concretes. Could you please discuss in the introduction this point? For instance: Carbon Dioxide Absorption by Blast-Furnace Slag Mortars in Function of the Curing Intensity. Energies 2019, 12(12), 2346; https://doi.org/10.3390/en12122346; Effect of curing time on granulated blast-furnace slag cement mortars carbonation. Cement and Concrete Composites 90 (2018) 257–265. https://doi.org/10.1016/j.cemconcomp.2018.04.006
Response: We have carefully cited these two articles and discussed in more detail the durability of concrete, especially the carbonation resistance.
9.Line 117: Another important aspect with regard to the durability is the Alkali-Silica Reaction (ASR). Could you please add the references on this topic?. For instance: Sustainable and Durable Performance of Pozzolanic Additions to Prevent Alkali-Silica Reaction (ASR) Promoted by Aggregates with Different Reaction Rates. Appl. Sci. 2020, 10, 9042. https://doi.org/10.3390/app10249042
Response: We have cited the article on the Alkali-Silica Reaction (ASR), and conducted a more detailed discussion in the revised manuscript.
10.Line 150: Could you please add the reference for the P•I 42.5 Portland cement?
Response: We have added the reference for the P•I 42.5 Portland cement in the revised manuscript.
11.Line 151: Could you please add the supplier of the cement and coal fly ash?
Response: We have added the supplier of the cement and coal fly ash in the revised manuscript.
12.Line 151: Why have you decided to keep constant the amount of coal fly ash (77 kg/m3)?
Response: According to our research group's previous experimental research on fly ash concrete and the reference to articles related to fly ash, we eventually replaced 15% of the cementitious material with fly ash, which is the optimal mixing amount to ensure that the strength of concrete mixed with fly ash will not be lower than that of ordinary concrete. Relevant articles were cited and added in the revised manuscript.
13.Line 154: Could you please add the type of liquid superplasticizer?
Response: We have added the type of liquid superplasticizer in the revised manuscript, which is a polycarboxylate superplasticizer.
14.Line 159: Particle size distribution (PSD) of the cement, coal fly ash and NS should be included in the investigation.
Response: We have added the particle size distribution (PSD) image of the cement, coal fly ash and NS in Figure 2 and Figure 3, respectively.
15.Line 159: A LOI of 5.24 could be considered high. Unburned carbon is an undesirable constituent of fly ashes to be utilized in the reinforced concrete construction. Therefore, it should be mentioned in the introduction. The problem is that the unburned carbon in fly ashes has several detrimental effects on the concrete. Especially, it increases the electrical conductivity of the concrete, changes the color of mortar and concrete (they may appear black), etc. Moreover, the water/(cement+fly ash) ratio, needed to obtain a cement paste with a required rheological properties or consistency, is higher for fly ashes with a high carbon content, increasing the corrosivity of metallic parts incorporated in the concrete. Finally, it causes a poor air entrainment behavior and mixture segregation. The following papers deal with this topic:
Freeman, E., Gao, Y-M., Hurt, R. and Suuberg, E.: 1997, Interactions of carbon-containing fly ash with commercial air-entraining admixtures for concrete, Fuel, 76, no. 8, 761–765. https://doi.org/10.1016/S0016-2361(96)00193-7
Ha, T.H., Muralidharan, S., Bae, J.H., Ha, Y.C., Lee, H.G., Park, K.W. and Kim, D.K.: 2005, Effect of unburnt carbon on the corrosion performance of fly ash cement mortar, Construction and Building Materials, 19, 509–515. https://doi.org/10.1016/j.conbuildmat.2005.01.005
Ehsan Ghafari, Seyedali Ghahari, Dimitri Feys, Kamal Khayat, Aasiyah Baig, Raissa Ferron. Admixture compatibility with natural supplementary cementitious materials, Cement and Concrete Composites, Volume 112, 2020, 103683, https://doi.org/10.1016/j.cemconcomp.2020.103683
Lim, W. Lee, H. Choo, C. Lee, Utilization of high carbon fly ash and copper slag in electrically conductive controlled low strength material, Construction and Building Materials, Volume 157, 2017, Pages 42-50, https://doi.org/10.1016/j.conbuildmat.2017.09.071
Response: Thank you for your suggestion. Based on the results in these articles, we have carefully discussed the effect of loss on ignition of coal fly ash on concrete. And the unburned carbon in coal fly ash does have harmful effects on concrete, and we have cited relevant articles and discussed this topic in depth.
16.Line 160: “pH” instead of “PH”.
Response: We have corrected this mistake in the revised manuscript.
17.Line 174: I agree that “The uniform dispersion of the NS is the key to ensuring the quality of a concrete”. Could you please add some references? For instance,
Effect of silica fume fineness on the improvement of Portland cement strength performance. CONSTRUCTION AND BUILDING MATERIAL Vol. 96, 2015, pp. 55–64. http://dx.doi.org/10.1016/j.conbuildmat.2015.07.092
Response: We have cited this article in the revised manuscript and added some discussion.
18.Line 199: What do you mean saying “In perchloride environments”. Perchloride is a chloride that contains more chlorine than other chlorides of the same element.
Response: What we originally wanted to express is: in an environment with a high concentration of chloride ions, the concrete structure will suffer damage. Sorry for the ambiguity caused by our improper statement, we have replaced "In perchloride environments" with “In chloride environments”.
19.Line 210: “… The RCM method proposed by Tang and Nilsson [49] was used in the chloride dif. …”. Therefore, the Chloride diffusion coefficient is calculated based on the guideline of NT BUILD 492? NORDTEST, “Chloride migration coefficient from non-steady-state migration experiments,” NT BUILD 492, 1999. The scope is ” This procedure is for determination of the chloride migration coefficient in concrete, mortar or cement-based repair materials from non-steady-state migration experiments.”. Then, I suggest to replace ” diffusion” by ” migration”.
Response: In fact, we use the rapid chloride ion migration (RCM) method in the chloride ion penetration test. The experimental principle of this test method is the rapid electromigration method, which is the same as NT BUILD 492. And we have replaced “diffusion” by “migration” in the revised manuscript.
20.Line 236: “3. Results and discussion”: In order to optimize the concrete mixture, you should follow a statistical procedure. For instance, check the following reference and discuss it in the introduction: Granulated Blast-Furnace Slag and Coal Fly Ash Ternary Portland Cements Optimization. Sustainability 2020, 12, 5783. https://doi.org/10.3390/su12145783
Response: We have cited this article in the revised manuscript and made some discussion.
21.Line 307: “… microstructure more denser … “ or “… microstructure denser … “?
Response: Sorry for the misunderstanding caused by our wrong statement, and we have corrected “… microstructure more denser … ” to “… microstructure denser …” in the revised manuscript.
22.Line 314: Agglomeration of NS has been reported in the literature: Combined effect of nano-SiO2 and nano-Fe2O3 on compressive strength, flexural strength, porosity and electrical resistivity in cement mortars. Mater. Construcc. Vol. 68, Issue 329, January–March 2018, e150. https://doi.org/10.3989/mc.2018.10716
Response: We have cited this article in the revised manuscript and added some analysis and discussion.
23.Line 393: Calcium hydroxide in solution reacts with NS. Then, why do you say “…crystals for reaction”?.
Response: This is our writing error. What we originally meant was that there was a lack of calcium hydroxide in the solution to react with NS to form a C-S-H gel. "crystals" has been deleted in the revised manuscript.
- Line 418: In “ … in the chloride diffusion coefficient, …”, I suggest to replace ”diffusion” by ” migration”.
Response: We have replaced diffusion by migration in the revised manuscript.
Reviewer 2 Report
- Could you specify the nature of the aggregates and the sand used?
- What type of superplasticizer did you use, please specify?
- Could you re-specify the surfaces and chemical compositions of cement and fly ash?
- Please add the standard deviations of your results on your graphs.
- Why not have made a composition simply with the cement without the fly ash?
- It is a shame not to have results on the porosity of the materials at different times. This would make it possible to determine the size of the pores and to fully understand the mechanisms of diffusion and mechanical resistance.
- A scanning electron microscope (SEM) analysis would allow us to better understand the hydrates formed by the combination of nanoparticles, fly ash and cement and to have a better interpretation of the results obtained.
Author Response
1.Could you specify the nature of the aggregates and the sand used?
Response: The nature of the aggregates and the sand were added in Table 4 and Table 5 in the revised manuscript.
2.What type of superplasticizer did you use, please specify?
Response: We have added the type of liquid superplasticizer in the revised manuscript, which is a polycarboxylate superplasticizer.
3.Could you re-specify the surfaces and chemical compositions of cement and fly ash?
Response: We have added the Composition of cement and fly ash in Table 3 in the revised manuscript.
4.Please add the standard deviations of your results on your graphs.
Response: The standard deviations was added in all graphs in the revised manuscript.
Among them, the impact energy in Figure 9 was calculated from the number of impacts, so the standard deviation was not added.
5.Why not have made a composition simply with the cement without the fly ash?
Response: In fact, as an industrial by-product, fly ash has been widely used in concrete in the last decades, and its recovery has brought significant economic and environmental benefits. And according to our research group's previous experimental research on fly ash concrete and the reference to articles related to fly ash, we eventually replaced 15% of the cementitious material with fly ash, which is the optimal mixing amount to ensure that the strength of concrete mixed with fly ash will not be lower than that of ordinary concrete. In the part of introduction, we introduced in detail the research significance and research progress of fly ash resource utilization.
6.A scanning electron microscope (SEM) analysis would allow us to better understand the hydrates formed by the combination of nanoparticles, fly ash and cement and to have a better interpretation of the results obtained.
Response: In the revised manuscript, a laser diffraction-type particle size analyzer was used to measure particle size distribution (PSD) of cement and coal fly ash, and a scanning electron microscope (SEM) was used to analys particle size of NS. PSD of coal fly ash and cement image was added in Figure 2, and SEM image of NS particles was added in Figure 3 in the revised manuscript.
Round 2
Reviewer 1 Report
Accept in present form.
Reviewer 2 Report
The authors have responded sincerely to all comments from critics and have edited the article based on their suggestions.